# In situ X-ray imaging of defect and molten pool dynamics in laser additive manufacturing

Chu Lun Alex Leung[1], Sebastian Marussi[1], Robert C. Atwood [2], Michael Towrie[3], Philip J. Withers[1] & Peter D. Lee [1]

The laser–matter interaction and solidification phenomena associated with laser additive manufacturing (LAM) remain unclear, slowing its process development and optimisation. Here, through in situ and operando high-speed synchrotron X-ray imaging, we reveal the underlying physical phenomena during the deposition of the first and second layer melt tracks. We show that the laser-induced gas/vapour jet promotes the formation of melt tracks and denuded zones via spattering (at a velocity of $1\,m\,s^{-1}$). We also uncover mechanisms of pore migration by Marangoni-driven flow (recirculating at a velocity of $0.4\,m\,s^{-1}$), pore dissolution and dispersion by laser re-melting. We develop a mechanism map for predicting the evolution of melt features, changes in melt track morphology from a continuous hemi-cylindrical track to disconnected beads with decreasing linear energy density and improved molten pool wetting with increasing laser power. Our results clarify aspects of the physics behind LAM, which are critical for its development.

[1] School of Materials, The University of Manchester, Oxford Road, Manchester M13 9PL, UK. [2] Diamond Light Source Ltd, Diamond House, Harwell Science & Innovation Campus, Didcot, Oxfordshire OX11 0DE, UK. [3] Central Laser Facility, Research Complex at Harwell, Science & Technology Facilities Council, Rutherford Appleton Laboratory, Didcot, Oxfordshire OX11 0QX, UK. Correspondence and requests for materials should be addressed to C.L.A.L. (email: alex.cl.leung@gmail.com) or to P.D.L. (email: pdlee123@gmail.com)

Laser additive manufacturing (LAM), including laser powder bed fusion (LPBF) and direct energy deposition (DED), fuses metallic, ceramic or other powders to build up complex 3D shapes, layer by layer. LAM has attracted significant interest in academia and industry.[1–5] The uptake of LAM in the production of safety-critical engineering structures[6], such as turbine blades[3,5], biomedical[7,8], and energy storage devices[9], is hindered by many technical challenges, including poor dimensional accuracy[10] and defects[11], e.g., lack of fusion[12], residual porosity[13–16] and spatter[17–20]. These defects can cause inconsistent mechanical performance (from yield stress to fatigue properties) of LAM components during service.[21–23] To reduce defect formation in LAM, a better understanding of the laser–matter interaction and powder consolidation mechanisms during LAM is required.

In LAM, the laser–matter interaction describes an event when the laser beam makes contact with the powder particles, molten pool, metal vapour, etc. Powder consolidation involves the fusion of powder particles into a solid bead by laser melting.[24] At present, the underlying mechanisms behind these processes are inadequately understood owing to the complex molten pool behaviour that occurs on very short time scales ($10^{-6}$–$10^{-3}$ s).[25–28] Many key phenomena have been partially revealed by real-time process monitoring devices in LAM machines[29,30], including spatter and line consolidation on the powder bed surface.[31] These in situ observations play a vital role in the development of computer simulations for LAM.[18,32] However, the hydrodynamic behaviour inside the molten pool, as melt-tracks form, has not been observed, hindering both our understanding of LAM and the development of process simulations tools.

Third-generation synchrotron radiation sources provide high flux X-ray beams enabling X-ray imaging with unprecedented temporal (tenths of microseconds) and spatial (a few micrometres) resolution.[33,34] This has been exploited to capture the dynamic molten pool behaviour during laser welding[35,36], as well as the formation and evolution of keyhole porosity within stationary laser-induced molten pools.[16]

Here, we perform in situ and operando synchrotron X-ray imaging of LAM to investigate and quantify the defect and molten pool dynamics. We reveal and elucidate the mechanisms by which a melt track, denuded zone, spatter, and porosity form during LAM, including pore migration, dissolution, dispersion, and bursting. The presented methods and results can enhance the understanding of additive manufacturing and other materials processing technology, such as welding and cladding, in which porosity and spatter are common issues.

## Results

**In situ and operando synchrotron X-ray imaging of LAM.** In order to capture thermophysical phenomena during LAM, we performed in situ and operando X-ray imaging on the I12: Joint Engineering, Environmental, and Processing (JEEP) beamline at Diamond Light Source (see Supplementary Fig. 1). The LAM process replicator can be accommodated on a synchrotron beamline and mimics an entry-level commercial LAM system, comprising a laser system, a loose powder bed with a packing density of 40%–60% and an inert processing environment. It enables a focused 1070 nm Ytterbium-doped fibre laser beam (continuous-wave (CW) mode, spot size of 50 µm, laser power ($P$) of 200 W, and scan speed ($v$) of 4 m s$^{-1}$, see Supplementary Fig. 2 and Supplementary Table 1) to be scanned across a powder bed (20 mm in width, 3 mm in height, and 0.3 mm in thickness, with an optional solid substrate) in a direction perpendicular to the X-ray beam. The powder bed is filled with gas atomised Invar 36 powder (Supplementary Fig. 3); a material of interest for

precision instruments, optical devices, electronic packaging, moulds, and aircraft tooling owing to its low coefficient of thermal expansion.[37,38] The powder bed is positioned inside an environmental chamber with X-ray windows and a flowing argon atmosphere. Here, we focus on examining the phenomena occurring during LAM on top of a loose powder instead of on a solid substrate; which is a geometry we refer to as the 'overhang condition' and is often encountered when building up complex 3D shapes.[39] The scan speed was selected to enable a continuous track to be formed during overhang conditions.

**Evolution of a single layer melt track during LAM.** We captured the evolution of a single layer melt track (MT1) during LAM (see Fig. 1 and Supplementary Movie 1). The contrast between the powder, molten pool, and melt track of Invar 36 in Fig. 1 correlates to their effective density with respect to the X-ray path length. The higher the effective density of the object the higher the X-ray attenuation and the darker it appears in the radiograph. The Invar 36 powder appears as light grey, whilst the molten pool and melt track are dark grey because their effective density is almost twice as dense as the powder, and hence attenuate more X-rays. Fig. 1a shows the changes in melt track morphology at the onset, middle, and final stages of LAM. At time $t = 0$, the laser beam ($P = 209$ W and $v = 13$ mm s$^{-1}$) switches on. It scans from right to left across the powder bed and consolidates powder particles into a molten pool ($t = 2.8$ ms), and subsequently evolves into a melt track which extends towards the bottom of the powder bed ($t = 338$ ms). As the melt track cools, it bends upwards, followed by pore formation in the last solidified region of the melt track ($t = 400$ ms). The four red dotted boxes and their magnified views highlight the evolution of powder consolidation (Fig. 1b–d), spatter (Fig. 1e), and porosity (Fig. 1f) during LAM.

During the initial stages of LAM (Fig. 1b), a molten pool forms *ca.* 100 µm (twice the laser spot size) below the powder bed surface ($t = 1.8$ ms) and rapidly grows into a 250 µm diameter sphere ($t = 2.8$ ms). Its growth rate slows as the equivalent diameter reaches *ca.* 400 µm (Fig. 1c), because powder spatter removes a significant amount of powder particles ahead of the laser beam[17] forming a powder-free (or denuded) zone in front of the molten pool[40], so that there are fewer powder particles available for subsequent powder consolidation. The laser beam moves ahead and forms a new molten pool further along the scan path ($t = 7.2$ ms). The growth rate of this newly formed molten pool is faster than $v$, thus the laser beam heats the molten pool whilst lowering its surface tension. Consequently, the newly formed molten pool coalesces with the first melt bead via wetting ($t = 7.4$ ms), revealing a key track growth mechanism. The combination of Marangoni convection inside the molten pool and the inward gas flow above the molten pool causes the molten pool to oscillate throughout LAM (Supplementary Movie 1).

The intense laser beam (power density of ~$10^6$ W/cm$^2$) causes metal vaporisation at the molten pool surface and generates a recoil pressure in the laser–matter interaction zone, resulting in a gas/vapour jet. We hypothesise that the hot powder and metal vapour heat the argon gas and cause a rapid gas expansion, resulting in a vapour jet that expands radially upwards at high-speed. This laser-induced gas/vapour jet entrains powder particles into a melt track which is a key track growth mechanism (purple dotted circles in Fig. 1d). It also induces powder spatter from the laser–matter interaction zone (orange dotted ellipse, Fig. 1e), forming a denuded zone. Unlike the droplet spatter mechanisms reported previously,[17,19,40] where spatter is ejected from the melt track, we observed that the majority of the droplet spatter during LAM in the overhang condition originated from the molten pools

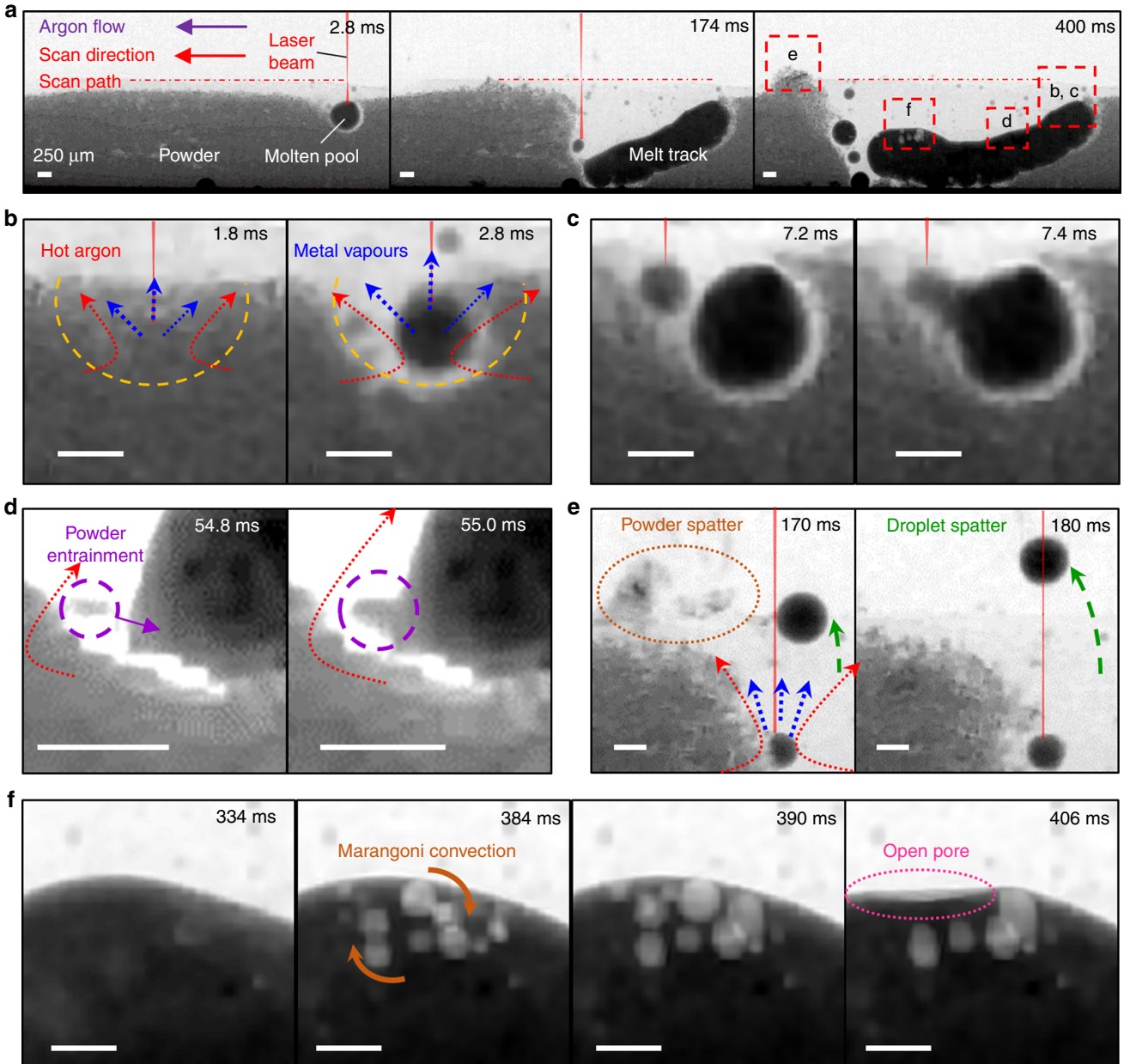

**Fig. 1** Time-series radiographs acquired during LAM of an Invar 36 single layer melt track (MT1) under $P = 209$ W, $v = 13$ mm s$^{-1}$ and LED = 16.1 J mm$^{-1}$ (see Supplementary Movie 1). **a** The melt track morphology at three key stages of LAM. **b** The formation of a molten pool and a denuded zone (yellow dotted line). The laser beam causes metal vaporisation, generating a recoil pressure at the interaction zone (blue dotted arrows) whilst indirectly heating up the surrounding argon gas (red arrows). The molten pool/track grows while enlarging the denuded zone by **c** molten pool wetting and **d** vapour-driven powder entrainment (orange dotted semi ellipse) which can lead to the formation of **e** powder spatter (purple dotted circle) and droplet spatter (its trajectory path is indicated by the green arrows). After the laser switches off at $t = 334$ ms, **f** pores nucleate, coalesce and collapse, resulting in an open pore (pink dotted line). All scale bars = 250 µm

ahead of the melt track, or via melting of powder spatter by the laser beam[41] (Supplementary Fig. 4a). We can distinguish whether spatter is hot or cold[41] by its greyscale values and diameter. Hot ejection has a diameter greater than 40 µm (or ~6 pixels) and it appears as black in the X-ray images as compared to cold powder agglomerates (which are less dense than the hot ejection and hence a dark grey colour in the X-ray images). Our results reveal that, in addition to the individual particle ejection[41], cold powder agglomerates are also ejected, interacting with the laser beam to become molten, transforming cold ejection into hot spatter. Although neither studies have a direct temperature measurement, our results support and build on the hypothesis of

Ly et al.[41] on the formation of cold and hot ejections during LAM.

The droplet spatter ejects vertically ($t = 170$ ms in Fig. 1e) and it follows the flow direction of the argon gas ($t = 180$ ms in Fig. 1e). We also reveal a less common phenomenon, a laser-induced gas expansion in droplet spatter (Supplementary Fig. 4b). Our observations on spatter formation under the overhang condition complement those of Ly et al.[41] on spatter formation during deposition on a solid substrate. These images explain why it has been difficult to produce a horizontal overhang feature, because the melt track formed by a slow laser scan speed promotes spatter formation which forces the melt track to extend

towards the bottom of the powder bed, preventing accurate builds.

It is evident that the volume of material added to the melt track by molten pool wetting exceeds the volume of material added to the melt track by vapour-driven powder entrainment, suggesting that molten pool wetting is the primary track growth mechanism of the main track for these conditions. The melt track continues to grow until the laser switches off at 334 ms. At this time, no porosity is evident in the melt track and spattering has also stopped (Fig. 1f). As the melt track cools, the gas solubility reduces significantly in the liquid metal[42,43], forming gas pores (e.g., hydrogen[44] or nitrogen[45,46]) ahead of the solid/liquid interface ($t = 384$ ms). These gas pores possibly originated from the gas porosity in the powder[47] or from moisture on the powder surface (or inside the environmental chamber), which can easily transfer into the molten pool during LAM. Our results show that most pores are swept outwards and downwards along the solidification front by the Marangoni convection, before rising in the middle of the pool and recirculating with velocities up to 0.4 m s$^{-1}$. As solidification progresses, the melt flow velocity reduces, the buoyancy forces exerted by the pores dominate,

causing the pores to reside near the track surface (Supplementary Movies 1 and 7) with some pores coalescing to form larger pores ($t = 390$ ms). Towards the end of the solidification, some pores adjacent to the top surface escape into the atmosphere by pore bursting, leaving a depression at the melt track surface (orange dotted circle, $t = 406$ ms and pink dotted ellipse). This explains how the open surface porosity observed by Qiu et al.[14] was formed. They hypothesised that the formation of open pores was due to incomplete melting or insufficient liquid feeding; however, we reveal that they were formed by a pore bursting mechanism.

**Influence of process parameters on melt track evolution.** To gain a better understanding of LAM, we conducted a systematic set of 15 trial runs to investigate the effects of laser power ($P$), scan speed ($v$), or linear energy density (LED = $P/v$)[48,49] on the evolution of the molten pool dynamics (see Fig. 2). Each trial uses a laser power density (~$10^6$ W/cm$^2$) sufficient to form molten pools. In order to visualise the progress of melt features over time, for each LED condition, the resulting time-series radiographs are transformed into a single time-integrated image by overlaying the

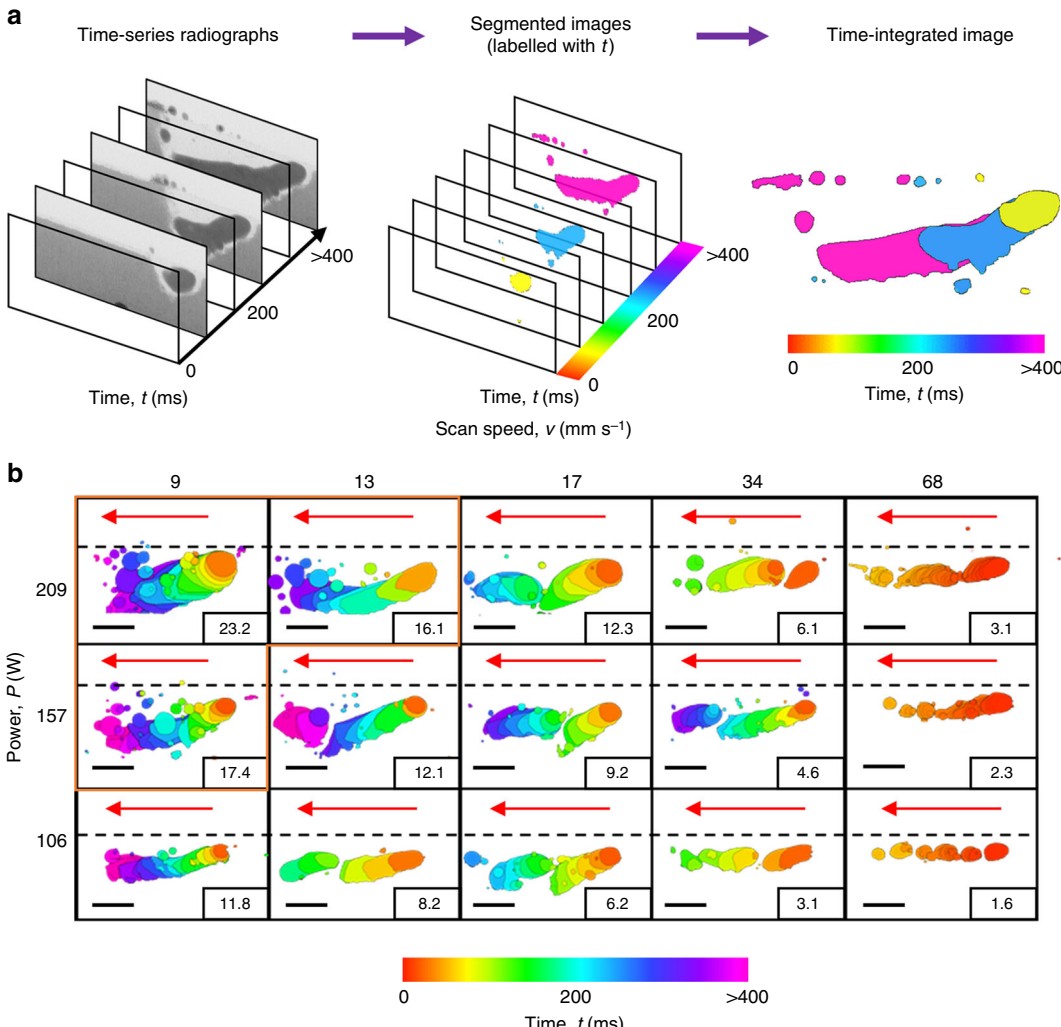

**Fig. 2** Time-resolved melt track morphologies under different process parameters. **a** Steps to produce a time-integrated image. Each time-series of radiographs is segmented and labelled with time, and then flattened into a time-integrated image. **b** A mechanism map combining 15 time-integrated images illustrates the dynamic changes of molten pool behaviour and melt track morphologies with respect to $P$, $v$, and LED (indicated by the numeric value at the corner of each time-integrated image). The black dotted line indicates the powder bed surface. Red arrows indicate the scan direction of the laser beam. Scale bars = 2 mm

time coloured segmented tracks on top of each other in reverse chronological order (Fig. 2a). We form a mechanism map by combining these time-integrated images with respect to their process parameters (Fig. 2b). Unlike the traditional $P$–$v$ process map[50] which shows only the final melt track morphology obtained from post-mortem analysis[51], Fig. 2b shows the dynamic changes occurring in the molten pool and also reveals the underlying phenomena associated with 15 different combinations of $P$ and $v$.

At constant $P$, the morphology of the melt track undergoes two transitions as $v$ increases: firstly from a continuous hemicylindrical track (Supplementary Movie 1) to two or more discontinuous hemi-cylindrical tracks (Supplementary Movie 2), and secondly from discontinuous hemi-cylindrical tracks to a series of disconnected metal beads, i.e., balling[52] (Supplementary Movie 3). SEM images of the track morphologies are shown in Supplementary Fig. 5.

Increasing $v$ reduces the laser energy transferred to the powder particles, thereby reducing the molten pool size and its peak temperature. When the molten pool temperature is reduced, its surface tension increases, hindering the wetting process and coalesce of the leading melt beads into the main melt track, eventually forming a discontinuous track. Upon increasing $v$ further (i.e., reducing LED to $< 4\,\mathrm{J\,mm^{-1}}$), the laser beam now moves faster than the growth rate of the molten pool so that the liquid metal immediately curls up into a sphere to minimise its surface energy. Spheriodisation of metal beads repeats itself until the laser beam switches off, resulting in balling (see Supplementary Movie 3). For overhang conditions, balling is primarily due to a high surface tension and leads to the formation of disconnected metal beads, contrary to the balling phenomenon, which is induced by a Plateau-Rayleigh instability, during LAM on a solid substrate.[20]

Increasing $P$ delivers more laser energy for powder consolidation while improving molten pool wetting; reducing $v$ increases the laser–matter interaction time. Both effects increase the likelihood of forming a continuous hemi-cylindrical track by increasing the molten pool temperature which reduces its surface tension and promotes molten pool wetting. In contrast, the movement of the liquid metal becomes less violent with increasing $v$ and decreasing $P$ (i.e., decreasing LED).

Furthermore, Fig. 2b shows that melt tracks exhibit similar morphology when processing at constant LED (e.g., $3.1\,\mathrm{J\,mm^{-1}}$).

For overhang conditions, the minimum LED required to form a continuous Invar 36 melt track is *ca.* $16\,\mathrm{J\,mm^{-1}}$, whereas the minimum LED to make Invar 36 parts on a solid substrate only requires a LED of $0.2\,\mathrm{J\,mm^{-1}}$.[37] The large difference in LED, because the effective thermal conductivity of the powder bed (present study) is lower than that of building on a solid starting block.[53] In addition, the wettability of the molten pool is significantly lower when building on powder support than building on a solid starting block or a prior solid track.[54] For both reasons, a slower $v$ is required to form a continuous track on power support as compared to solid; therefore, requiring us to use different processing conditions to those used by Qiu et al.[37]. The mechanism map (Fig. 2) reveals morphological transitions of the melt track similar to reference studies[20,37,54], suggesting that the processing regime we selected is reasonable for laser melting directly on powder support.

We have selected three continuous tracks (highlighted by the orange outline in Fig. 2b) and quantified the changes in the melt track geometry over time. The track lengths are found to be $19.3\% \pm 0.3\%$ longer than the nominal scanned length (4 mm). The analysis shows that these melt tracks have undergone 3%–5% solidification shrinkage (see details in Supplementary Fig. 6, Supplementary Table 1 and Supplementary Movie 4). This analysis highlights a build accuracy issue in LAM which is closely linked to the process parameters[10], e.g., increasing $P$ and LED increase the melt depth. These real-time measurements of the molten pool geometry can be used for verifying existing computer simulation tools for the prediction of build accuracy in LAM.

**Melt track evolution during layer-wise LAM.** While much can be learned from observing the laser–matter interaction in the overhang condition, it is a common practice in LAM to build a part by adding a melt track on top of another track. Hence, we added a second powder layer on MT1 and perform another trial under identical process conditions to MT1 to form a second layer melt track (MT2). Our results reveal the interactions between MT1 and MT2 during LAM (Fig. 3a–d and Supplementary Movie 5) and uncover three pore formation mechanisms (Fig. 3i–k).

The initial powder consolidation phenomena in MT2 ($<5\,\mathrm{ms}$) are similar to those observed in MT1, in which the laser creates a small denuded zone while forming a molten pool just below the powder level (Fig. 3a). The molten pool grows until it reaches a

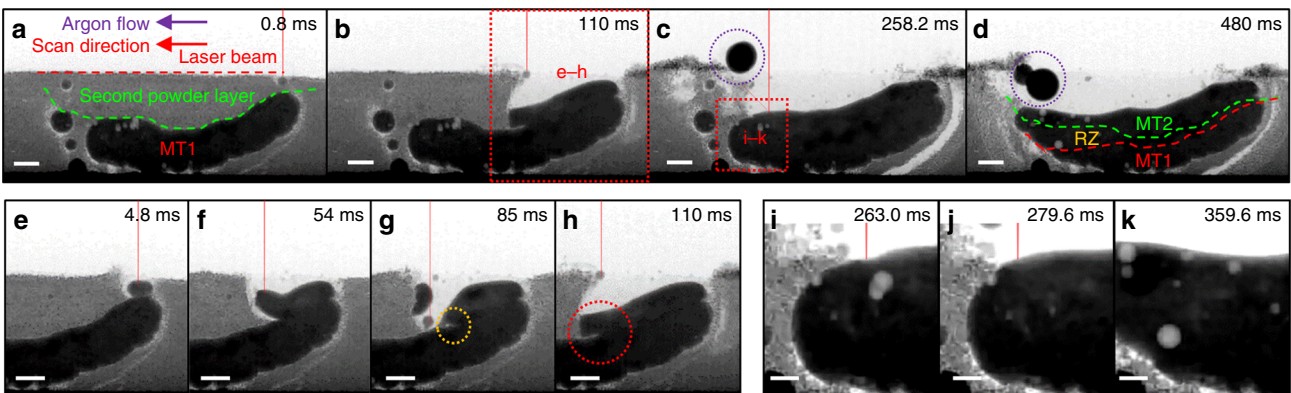

**Fig. 3** Time-series radiographs acquired during LAM of an Invar 36 s layer melt track (MT2) under $P = 209\,\mathrm{W}$, $v = 13\,\mathrm{mm\,s^{-1}}$ and LED $= 16.1\,\mathrm{J\,mm^{-1}}$, see details in Supplementary Movie 5. Snapshots of MT2 at various stages of LAM: **a** formation of a molten pool on the second powder layer; **b** the growth of MT2 via wetting on MT1; **c** MT2 undergoes laser-re-melting, where RZ indicates the laser re-melting zone; **d**, MT2 completely solidifies. (**c-d**) The large droplet spatter (purple dotted circle) failed to eject from the powder bed. Other key phenomena involved during molten pool wetting: **e** re-melting of MT1; **f** an extension of MT2 before wetting on MT1; **g** entrapment of gas bubble (see yellow dotted circles); and **h** thin powder layer (see red dotted circles). Three pore evolution mechanisms induced by laser re-melting: **i** pore coalescence; **j** pore dissolution; and **k** pore dispersion into smaller pores. Scale bars = 500 μm

diameter of *ca.* 300 μm, over half the diameter of that in MT1 (*ca.* 500 μm). Then the molten pool wets onto the surface of MT1 (Fig. 3e), causing it to undergo re-melting (see greyscale changes in Supplementary Movie 5), which is the mechanism by which a second (or *n*th) layer forms. This shows how the underlying layer facilitates track growth via molten pool wetting and explains why it is much easier to form a melt track on a solid substrate than in the overhang condition.

In addition to track growth, molten pool wetting also plays a key role in pore formation and growth (see Fig. 3e–h and Supplementary Movie 5). Marangoni-driven melt flow causes the liquid metal (with a high surface tension) to fold over the surface of MT1 (Fig. 3f), entraining an argon bubble[18,55] (yellow dotted circle in Fig. 3g). Since the argon gas solubility is <0.1 part per billion in Fe-based alloys[56,57], argon will remain in the bubble unless it reaches the surface and is released into the atmosphere. The Marangoni-driven flow also entrains a thin powder layer between MT1 and MT2, as indicated by the abrupt changes in the image contrast at the centre of Fig. 3h. If the thin powder layer remains in between the melt tracks, this could lead to the formation of lack of fusion defects and interlayer pores.

In contrast to the MT1 trial, porosity nucleates and grows throughout LAM of MT2 rather than only after the laser switches off (Supplementary Movie 5). We hypothesise that MT1 acts as a heat sink, conducting a significant amount of heat energy from the laser–matter interaction zone, causing a rapid solidification of MT2 and producing gas pores. The large temperature gradient in MT2 causes large variations in surface energy and concomitant Marangoni forces, resulting in a fast melt flow, swirling the pores inside the melt track and along the laser scanning direction. This removes many interlayer pores by allowing them to flow towards the surface and escape from the melt track.

In addition to the reported pore growth mechanism, we also see pore dissolution and dispersion mechanisms occurring during LAM of MT2 (Fig. 3c, i–k). At the end of MT2, spatter has removed many powder particles above MT1 (purple dotted circle Fig. 3d). Therefore, the laser beam re-melts a large portion of MT1 (see the re-melting zone (RZ) in Fig. 3d). Laser re-melting promotes existing pores in MT1 to coalesce and grow at the expense of others (Fig. 3i), reducing the pore density. Assuming that the pores contain hydrogen or nitrogen, the gas solubility increases with temperature, causing gas pores to dissolve back into the melt (Fig. 3j). After the laser switches off, small pores reappear in MT2 during cooling as the solubility reduces (Fig. 3k); suggesting laser re-melting disperses large pores into smaller ones rather than eliminating them.

In the MT2 experiment, the droplet spatter fails to eject completely, it grows via powder amalgamation while rolling along the top surface of the BN walls (purple dotted circle, Fig. 3c). This phenomenon is unlikely to carry forward to a commercial AM process because the droplet spatter is expected to redeposit onto a melt track. Due to the thin powder bed, the melt track sometimes is in contact with the BN walls, restricting the hot argon gas flow inside the powder bed, resulting in powder spatter at both sides of the melt track (Fig. 3c, d and Supplementary Movie 5), but does not alter the underlying physics, as discussed in the Methods section. The walls also accentuate the side ejection of powder

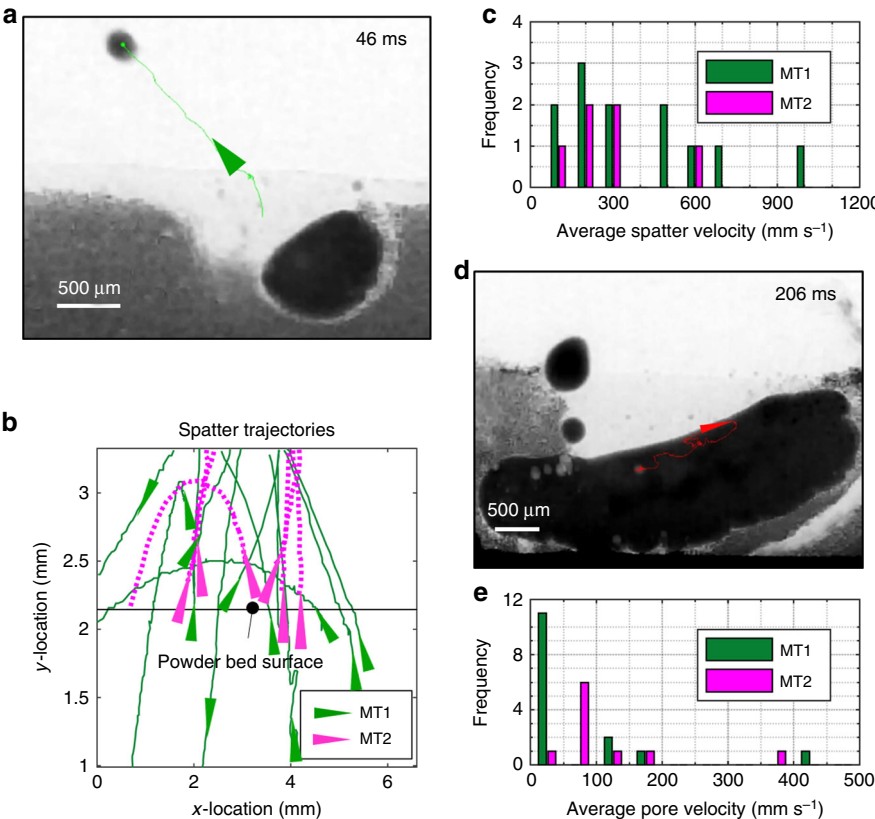

**Fig. 4** Tracking the movement of spatter and pore during LAM of the two melt tracks (MT1 and MT2). **a** An example of a spatter trajectory overlaid on a radiograph (see the spatter tracking video in Supplementary Movie 6). **b** An overlay image contains all the spatter trajectories. **c** A histogram of the average spatter velocities. **d** An example of a pore flow trajectory overlays on a radiograph. The centrifugal Marangoni-driven flow forces the pore to flow in a clockwise direction (see the pore tracking video in Supplementary Movie 7). **e** A histogram of the average pore flow velocities

(Supplementary Movie 5), further supporting the hypothesis that argon gas expansion contributes to spatter formation.

**Time-resolved defect quantification.** To reveal the underlying defect formation and growth mechanisms, as well as the hydrodynamic behaviour of the melt fluid, we track the movements of droplet spatter and pores during the evolution of MT1 and MT2 (Supplementary Movies 6 and 7). We overlay a typical spatter trajectory path on a radiograph (Fig. 4a) to illustrate how the spatter is tracked. Due to the spatial resolution of the radiographs, we are only able to track spatter having a size range of 40–350 μm. The spatter trajectories predominately eject in the direction of gas flow and a scanning laser beam (Fig. 4a).[58] During the formation of MT1, droplet spatter usually originates near the denuded zone (Fig. 4b). Here, the laser beam heats a large powder volume with a large surface area, inducing metal vaporisation while indirectly heating the argon, creating a strong jet that causes powder and droplet spatter to move at high velocities (Fig. 4c). During the formation of MT2, the laser beam fuses the powder layer with MT1 so that much of the heat energy is transferred to MT1, and hence only a small amount of energy contributes to gas/vapour jet and spatter formation. The majority of droplet spatter in MT2 forms as the powder spatter passes through the laser beam (100–500 μm) above the powder bed surface, this confirms that the hot argon gas assists the formation of droplet spatter. Additionally, our observation shows that droplet spatter is more likely to form under overhang conditions than LAM on a solid substrate (Fig. 4c).

The flow direction and velocities of liquid metal can be determined by tracking the pore motion. A tracked pore trajectory verifies the expected dominance of centrifugal Marangoni convection (Fig. 4d and Supplementary Movie 7).[59] The melt zone in MT2 is much smaller than that in MT1 (Fig. 3d), because most of the heat energy conducts towards MT1, resulting in a high thermal gradient at the melt zone of MT2. The liquid metal flows three times faster in MT2 than MT1 as shown by the modal average pore velocity, which is 75 mm s$^{-1}$ in MT2 compared to 25 mm s$^{-1}$ in MT1 (Fig. 4e).

In summary, we have used synchrotron X-ray imaging to uncover key mechanisms of laser–matter interaction and powder consolidation in situ and operando during LAM, including the formation and evolution of the melt tracks, porosity, spatter, and denuded zone. Further, the time-resolved quantification of the pore and spatter movements give crucial information about their velocities and direction, not possible to acquire using other techniques. We have demonstrated that Marangoni convection dominates the liquid metal flow, reaching a velocity of 400 mm s$^{-1}$. Our mechanism map provides additional insight into the dynamic changes in melt track behaviour across a range of process parameters, e.g., improved molten pool wetting with increasing $P$, and the morphological transition from first a continuous to an interrupted hemi-cylindrical melt track and a series of independent molten beads as $v$ increases. The methodology introduced here also sheds light on the mechanisms of pore formation, including the migration, dissolution, dispersion, and bursting of pores during LAM. Future investigations in these areas will deepen our fundamental understanding of the nature of the laser–matter interaction.

## Methods
**Material characterisation.** The gas atomised Invar 36 powder (TLS Technik GmbH & Co. Spezialpulver KG, Germany) was characterised using scanning electron microscopy (SEM) JEOL JSM-6610LV (Tokyo, Japan) and analysed by Energy-Dispersive Spectroscopy (EDS). Example SEM images are shown in Supplementary Fig. 3a. We performed an elemental analysis using ANALYZER (Aztec, Oxford Instrument plc, UK) in the same field of view. The elemental composition of the powder is shown in Supplementary Fig. 3b and Table 1. The oxygen

**Table 1 Composition analysis of Invar 36 powder using SEM-EDS**

| Element | Ni | Fe | Al | P | S | O |
|---|---|---|---|---|---|---|
| Weight (%) | 29 | 65.3 | 0.1 | 0.1 | 0.3 | 5.3 |

concentration of 5.3 weight percent (wt.%) indicated that the as-received powder had an oxidised surface. We segmented the particles using Otsu's method[60] and then separated them using a watershed algorithm implemented in the imaging toolbox of MATLAB 2016A (MathWorks, USA). Based on the quantitative analysis of SEM images, the particle size distribution was found to be in the range of 5–70 μm with a $d_{50}$ of 16 μm, see Supplementary Fig. 3c and 3d.

**In situ and operando synchrotron X-ray imaging.** To reveal the sequential thermophysical phenomena arising during LAM, we used the hard X-ray beamline I12: Joint Engineering Environmental, and Processing (JEEP) at Diamond Light Source, UK for the synchrotron X-ray radiography experiments. The LAM process replicator (LAMPR) was mounted onto a sample stage in I12: JEEP (Supplementary Fig. 1) which consists of a 200 W Ytterbium-doped fibre laser (wavelength of 1070 nm, TEM$_{00}$, continuous wave (CW) mode) (SPI Lasers Ltd, UK), a loose powder bed with a packing density of 40%–60%, an environmental chamber, IR reflective optics and a laser safety enclosure (Supplementary Fig. 1a). The laser beam was directed through a collimator, a beam expander, and an $X$–$Y$ galvanometer scanner (Laser control systems Ltd., UK) coupled with f-theta scan lens to focus its spot size down to *ca.* 50 μm at the powder bed surface (Supplementary Fig. 1b). A full laser beam characteristic is shown in Supplementary Table 2.

A sample holder was positioned at the centre of the environmental chamber that securely held a powder bed of 30 mm × 3 mm × 0.3 mm (width × length × thickness) made of a sandwiched structure of boron nitride (BN) plates (Supplementary Fig. 1c). In each experiment, loose powder particles of Invar 36 were loaded into the cavity of the powder bed by mechanical vibrations. Then, the sample holder was inserted into the environmental chamber from its side port. Once the environmental chamber was completely sealed, it was backfilled with argon gas at a flow rate of 4 l min$^{-1}$ to reduce oxidation and metallic plume adsorption during LAM. Next, the laser beam was scanned over a 4 mm line across the powder bed at five different scan speeds (9, 13, 17, 34, and 68 mm s$^{-1}$) and three laser powers (106, 157, and 209 W). The reported $P$ and $v$ values were calibrated after the in situ experiments (see details in Supplementary Fig. 1).

Although the laser beam (50 μm) does not interact with the BN walls (which are 125 μm away), the thermal field it creates will interact with these walls by heat conduction through the low effective thermal conductive powder. However, prior studies have demonstrated thin wall radiography correctly captures solidification physics in a range of processes from semi-solid processing[61] to laser welding[62]. In addition, prior work on defect formation has been scaled to full industrial processes and used to design automotive components for many years, demonstrating the capture of process relevant physics, including Marangoni-driven flow and pore formation[63]. Boron nitride (BN) was selected for the walls as it was used in many prior solidification studies (with interaction times up to 3 orders of magnitude greater than in LAM) because of its non-wetting and low X-ray attenuation properties, e.g., for radiography[42] and tomography[64].

All the in situ experiments were observed using a monochromatic X-ray beam at 55 keV and a high-speed X-ray imaging system comprising a CMOS camera with a 12 GB high-speed internal memory (Miro 310 M, Vision Research, US) coupled with module 3 custom-made optics (I12: JEEP, DLS, UK).[39] This provided a field of view (FOV) of 8.4 mm × 3.3 mm (width × height) and a pixel resolution of 6.6 μm. The image acquisition was synchronised with the LAMPR using a ring buffer mode that continued to record images into the internal memory of the camera until the laser was triggered. Before the laser trigger point, 100 images were recorded as flat field images, and then a further 100 frames of dark field images were taken without switching on the X-ray beam. These two sets of images were taken for flat field correction to remove image artefacts caused by pixel variations and thermal counts during image acquisition. Once the laser was fired, a series of radiographs was captured at 5100 fps (with an exposure time of 196 μs) using a 700 μm thick LuAg: Ce scintillator.

**Image processing and quantification.** We processed all the acquired radiographs using MATLAB 2016a. Firstly, we applied a flat field correction using the following equation: FFC = (I$_0$−Flat$_{avg}$) / (Flat$_{avg}$−Dark$_{avg}$), where FFC is the flat field corrected image, I$_0$ is the raw image, Flat$_{avg}$ is the average of 100 flat field images and Dark$_{avg}$ is the average of 100 dark field images. These FCC images were denoised by VBM3D[65], followed by a custom background subtraction to remove most of the non-moving objects. After that, the molten pool and droplet spatter were segmented using Otsu's threshold method[60]. The molten pool geometries (length, width, and area) were quantified by standard MATLAB built-in functions. The area shrinkage (%) of the molten pool is calculated based on: (maximum molten pool

area at the onset of the cooling stage − molten pool area after cooling) divided by (the molten pool area after cooling) × 100%. Furthermore, we used a manual tracking plugin from ImageJ to track the movement of the droplet spatter and pores in MT1 and MT2. The distance and average velocity were computed based on the pixel locations obtained from the individual trajectory path and duration of each event.

**Data availability**. Representative samples of the research data are given in the figures (and supplementary data). Other datasets generated and/or analysed during this study are not publicly available due to their large size but are available from the corresponding author on reasonable request.

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

## Acknowledgements

The authors acknowledge financial support from the AMAZE project (Additive Manufacturing Aiming towards Zero Waste and Efficient Production of High-Tech Metal Products) funded by the seventh Framework Programme of the European Commission (contract FP7-2012-NMP-ICT-FoF-313781), and the EPSRC (EP/I02249X/1, EP/P006566/1, and EP/M009688/1). We also acknowledge the use of facilities and support provided by the Research Complex at Harwell and thank the Diamond Light Source for providing the beamtime (EE11761-1) and staff at JEEP-I12 beamline for technical assistance. Many thanks to team members from the Lee group and MXIF for their assistance in this beamtime: Dr. Anne-Laure Fauchille, Dr. Chris Simpson, and Dr. David Eastwood.

## Author contributions

P.D.L. and C.L.A.L. conceived the project. C.L.A.L., S.M., and M.T. led the design of the laser additive manufacturing replicator. C.L.A.L. designed and performed the experiments, with all authors contributing. C.L.A.L. performed the data analysis. C.L.A.L. and P.D.L. led the results interpretation and paper writing, with all authors contributing.

## Additional information

**Competing interests:** The authors declare no competing interests.

