## [Peer Review File · Nature Communications]

Reviewers' Comments:

Reviewer #1:

None

Reviewer #2:

Remarks to the Author:

The paper by Leung et.al. demonstrates how high speed radiography can be used to probe, in an operando manner, the laser material interaction that occur during laser powder bed fusion processes. The paper is the first report of data produced by a laser additive manufacture system coupled with the x-ray imaging facility at JEEP beamline at the Diamond Light Source.

Although this is a nice paper it is not at the Nature Communications level. It's an instrumentation paper that's stretching to represent itself as uncovering new physics, which it really doesn't do. Some of the splatter mechanisms that are described as novel are reported in a recent paper by Sonny Ly et al. Metal vapor micro-jet controls material redistribution in laser powder bed fusion additive manufacturing, Scientific Reports (2017). DOI: 10.1038/s41598-017-04237-z , which they fail to cite.

The biggest issue with paper is that their results are being effected by interaction with the sample holder walls. The sample is only 300 microns thick, and the power/speed combinations used, especially in a deep powder bed with no substrate, are definitely producing a melt pool big enough that it's thermally interacting with the boron nitrite windows, perhaps even in physical in contact with them. This fact complicates any claims that they are observing process-relevant physics.

That said, this is very encouraging commissioning data but this paper belongs in a AM journal, RSI or a similar journal, not Nature communications.

Reviewer #3:

Remarks to the Author:

This manuscript reports important new observations on the physics of melt pool and defect formation in laser-based additive manufacturing. This is possible through the development of innovative time-resolved synchrotron imaging approaches and subsequent analysis of large volumes of data. The manuscript addresses a topic of broad interest to the scientific and engineering community and employs insightful experimental approaches that uncover new physics associated with laser-powder interaction, melting and defect formation. The reviewer recommends that this should be published after the authors address the points below.

1. In the abstract the authors report "changes in track morphology from a line to spheres with decreasing linear energy density". This is confusing – do the authors mean to point out a transition in melt pool shape from hemispherical to more ellipsoidal? How does linear energy density apply to a single hemispherical spot?

2. A reader not intimately familiar with additive manufacturing may not understand the significance of "overhang conditions". This should be explained in a bit more detail.

3. Line 52: Define JEEP or leave the acronym to the supporting online materials

4. Line 70: What is the definition of "powder consolidation"?

5. Line 72: What is meant by "many other mechanisms"?

6. Figure 1:

Explain the contrast mechanisms associated with the dark contrast melt pool and the size of the initial and steady state melt pool diameter relative to the focused diameter of the beam.

What is the definition of the "denuded zone"?

Why have the authors not considered re-deposition of spatter and droplets as a key aspect of the process? The images at 384 and 390ms seem to suggest this possibility.

7. Line 87: "The powder particles are often entrained by the gas/vapour jet": How does one determine whether or not powder particle is entrained?

8. Line 106: If powder spatter is remelted by the laser beam, why wouldn't individual powder particles also be remelted? Is this a resolution issue? The authors should comment on resolution issues somewhere in the main body of the paper.

9. Figure 2: Clearly identify scan direction and include a scale on the times in the figure.

10. Figure 3 – Explain the contrast and shape of MT1 (why is there a dark ellipsoidal shape for MT1?), how is it clear that MT2 is liquid in the first frame? Why is there an abrupt change in contrast in MT2 at the center in frame b? Where is the purple line in k?

11. Line 225: During the formation of MT2, the laser strikes the top surface of MT1 rather than the powder surface, with a greatly reduced in surface area, creating a gas/vapour jet that is only sufficient to eject powder particles." The authors are suggesting that the powder particles absorb no energy in MT2? This seems unlikely. Also, there is no estimate of the fraction of energy absorbed in the MT1 melting configuration as compared to MT2.

12. The observation of different porosity formation mechanisms in MT2 is particularly important. The assertions (Line 192) about thermal gradients, surface energy gradients and Marangoni flow should be clarified – to what extent are there observations/simulations to support this and to what extent are the authors hypothesizing?

13. Section 5: What is the minimum and maximum size of spatters being tracked in the trajectory analysis? Again, redeposition should be addressed in the context of beam velocity and redeposition time; is it just at a length scale and time scale beyond what is observed here?

14. Are there any sights on final surface roughness related to the phenomena observed?

15. Finally, the numerous grammatical errors in the manuscript lead the reviewer to question whether the two senior authors have thoroughly enough reviewed this manuscript. Some examples include:

Line 27: "Laser additive manufacturing.....have attracted"

Line 28: " build ups"

Line 34: "To solve these challenges require a better understanding...."

Line 37: "not well explained owing to complex molten pool behaviour occurs on very short time scales (10⁻⁶ - 10⁻³ s).

Line 42: "the development of LAM and simulation tools required to apply on new materials"

Line 76: "because spatter removes a large amount of materials"

Line 82: "Notably, the hydrodynamic movement of the melt fluid during wetting combined with the differential vapour pressure at the laser-powder interaction zone, causing the molten pool to oscillate throughout LAM, a similar prediction is reported in selective electron beam melting."

Line 94: "The existing hypothesis suggests "...what is the hypothesis?"

Reviewer #4:

Remarks to the Author:

Excellent, original and comprehensive study of the physics of the selective laser melting process. It clearly elucidates the physical processes occurring during laser melting of powder materials under a range of operating conditions. I was wondering if you can clarify what effect does powder compaction play in the process. In your experiments the powder was loosely spread on the substrate while in practice its packing density is greater which can impact the process. Also have found numerous grammatical expressions which should be corrected before publication. Some listed below.

Line 37 Include "that" after "behaviour"

Line 39 Change "These" to "This"

Line 76 and 77, I would change "material" to "powder"

line 83 Change "causing" to "causes"

Line 89 Sentence starting with "Similar ..." needs to be looked at.

Line 96 Change "attributes" to "contributes"

Line 121 Change "they" to "They" after "."

Line 213 Figure 3. Purple dotted circles not included on my copy of the manuscript.

Line 226 Delete "in" after "reduced"

Line 230 Change "confirms" to "confirming"

Detailed response to Reviewers' comments:

Our response is written in purple.

Reviewer #1 (Remarks to the Author):

No remarks to the author.

No response to review #1:

Reviewer #2 (Remarks to the Author):

The paper by Leung et.al. demonstrates how high speed radiography can be used to probe, in an operando manner, the laser material interaction that occur during laser powder bed fusion processes. The paper is the first report of data produced by a laser additive manufacture system coupled with the x-ray imaging facility at JEEP beamline at the Diamond Light Source.

Although this is a nice paper it is not at the Nature Communications level. It's an instrumentation paper that's stretching to represent itself as uncovering new physics, which it really doesn't do. Some of the splatter mechanisms that are described as novel are reported in a recent paper by Sonny Ly et al. Metal vapor micro-jet controls material redistribution in laser powder bed fusion additive manufacturing, Scientific Reports (2017). DOI: 10.1038/s41598-017-04237-z, which they fail to cite.

The biggest issue with paper is that their results are being effected by interaction with the sample holder walls. The sample is only 300 microns thick, and the power/speed combinations used, especially in a deep powder bed with no substrate, are definitely producing a melt pool big enough that it's thermally interacting with the boron nitrite windows, perhaps even in physical in contact with them. This fact complicates any claims that they are observing process-relevant physics. That said, this is very encouraging commissioning data but this paper belongs in a AM journal, RSI or a similar journal, not Nature communications.

Responses to review #2:

Thank you for your comments on our manuscript. We hope that in the light of our changes and response to your comments below that, in common with the other referees, you will consider the paper as being appropriate for publication in *Nature Communications*. We would also like to convince you that the focus of the paper is on revealing new mechanisms that have not been captured by imaging at the surface of molten pools / melt tracks, or inside a stationary molten pool, albeit revealed via using a unique *in situ* additive manufacturing process replicator. Via imaging inside the molten pool during laser additive manufacturing of melt tracks, our work complements previously existing hypotheses, adding the following new insights:

- The mechanisms by when melt tracks form under overhang condition
- Revealing the mechanisms of pore formation during AM, including migration, dissolution, dispersion and bursting.
- Capturing how Marangoni-driven flow entrains porosity, sweeping it along a melt track

- The first measurements of the flow direction and velocities of the liquid metal during AM
- First *in situ* observation of open pore formation and the mechanisms causing them.
- Elucidating new spatter formation mechanisms under different build conditions, including quantifying the direction and velocity of spatter for these conditions.

Re: “Some of the splatter mechanisms that are described as novel are reported in a recent paper by Sonny Ly et al.”

We are grateful for your pointing out that our observations confirm the spatter mechanism in the paper published by Ly *et al.* 2017 this summer, and it is now cited in the revised manuscript. Our results, performed in the overhang condition expand upon the spatter mechanisms reported by Ly *et al.* (based on building on a substrate). We identify additional spatter formation mechanisms, *e.g.* the ejection of powder which agglomerates as it passes through the laser beam well above the molten pool. We also explain why spatter is formed more frequently for additive manufacture under overhang conditions as compared to when metal is laid down onto previously deposited metal. Based on our X-ray images, we are able to distinguish the difference between powder and droplet spatter. We also report the previously unseen effect of laser-induced gas expansion inside droplet spatter.

RE: “The biggest issue with paper is that their results are being effected by interaction with the sample holder walls..... This fact complicates any claims that they are observing process-relevant physics”

We agree with the reviewer that there is a thermal interaction with the BN walls; especially for the low scan speeds used which are required to obtain continuous track in overhang condition. However, there are a wealth of papers demonstrating that thin wall radiography correctly captures solidification physics (*e.g.* Kawahito *et al.* 2004 for laser welding) and that this physics is "**process-relevant**". In particular, the work on defect formation has been scaled to full industrial processes and used to design automotive components for many years (Wang *et al.* 2010). These and other papers demonstrate that the thermal interaction does not alter the mechanisms by which porosity forms nor Marangoni-driven flow in the molten pool. With regards to the container being made of boron nitride (BN), this material is a typical containment material used in many of these prior solidification studies (*e.g.* Lee and Hunt 1997 for radiography of defects; Karagadde *et al.* 2015 for tomography) where the interaction times are up to 3 orders of magnitude longer. The correlation to Ly *et al.* also substantiates the results. We have now added details regarding the effects of interaction with the BN walls in the manuscript, and discussed the impact.

We feel that our manuscript provides important new insights, as listed above. We believe that the submitted manuscript appeals both researchers in AM, and to the broader readership of *Nature Communications*, such as those interest in the application of high frame rate X-ray imaging and image processing to study laser-

matter interactions, fluid dynamics, powder consolidation, solidification and defect formation.

Reviewer #3 (Remarks to the Author):

This manuscript reports important new observations on the physics of melt pool and defect formation in laser-based additive manufacturing. This is possible through the development of innovative time resolved synchrotron imaging approaches and subsequent analysis of large volumes of data. The manuscript addresses a topic of broad interest to the scientific and engineering community and employs insightful experimental approaches that uncover new physics associated with laser-powder interaction, melting and defect formation. The reviewer recommends that this should be published after the authors address the points below.

Response: We'd like to thank reviewer #3 for your constructive criticisms and are encouraged by your desire to see the paper published in *Nature Communications*. Here are our responses to your specific comments.

1. In the abstract the authors report “changes in track morphology from a line to spheres with decreasing linear energy density”. This is confusing – do the authors mean to point out a transition in melt pool shape from hemispherical to more ellipsoidal?

Response: If we understand the question correctly, we have now clarified the abstract to make the track morphology statement clearer.

Abstract [Line 20 - 23]: *“We develop a mechanism map for predicting the dynamic evolution of melt features, including: the change in melt track morphology from a continuous hemi-cylindrical track to disconnected beads as linear energy density is decreased; and improved molten pool wetting as laser power is increased.”*

How does linear energy density apply to a single hemispherical spot?

Response: “Linear energy density”, or LED, represents a ratio between the laser power and laser scan speed, it is a term used in the field to correlate the results with the process parameters. For example: the *P-v* study (**Fig. 2**), we performed 15 separate line scans using 15 different combinations of *P* and *v*. Therefore, the LED applies to all cases, including those line scans resulted in a series of disconnected beads via balling. We have clarified this in the revised manuscript (both in the abstract and main text).

2. A reader not intimately familiar with additive manufacturing may not understand the significance of “overhang conditions”. This should be explained in a bit more detail.

Response: We have added a sentence to explain the overhang condition to a more general readership, as below:

[Line 63 – 66]: *“Here, we focus on examining the phenomena occurring during LAM on top of a loose powder instead of on a solid substrate; which is a geometry we refer to as the ‘overhang condition’ and is often encountered when building up complex 3D shapes”*

3. Line 52: Define JEEP or leave the acronym to the supporting online materials
Response: We have now spelt out the acronym JEEP in full at its first occurrence.

[Line 52 - 53]: *“In order to capture the thermophysical phenomena during LAM, we performed in operando X-ray imaging using a combination of a LAMPR and I12: Joint Engineering Environment Processing (JEEP) beamline at Diamond Light Source.”*

4. Line 70: What is the definition of “powder consolidation”?

Response: In the context of the manuscript, the powder consolidation involves fusion of powder particles into a dense object by laser melting. We have clarified that in the revised manuscript:

[Line 36 - 37]: *“The powder consolidation involves the fusion of powder particles into a solid bead via laser melting.”*

5. Line 72: What is meant by “many other mechanisms”?

Response: We have clarified the different mechanisms in the revised manuscript:

[Line 79 – 81]: *“The four red dotted boxes and their magnified views highlight the evolution of powder consolidation (**Fig. 1b-d**), spatter (**Fig. 1e**) and porosity (**Fig. 1f**) during LAM.”*

6. Figure 1: Explain the contrast mechanisms associated with the dark contrast melt pool and the size of the initial and steady state melt pool diameter relative to the focused diameter of the beam.

Response: In the revised manuscript, we have explained that the contrast mechanism is related to the X-ray absorptivity of the powder particles, molten pool and melt track:

[Line 70 - 74] *“The contrast between the powder, molten pool and melt track of Invar 36 in Fig. 1 correlates to their effective density with respect to the X-ray path length. The higher the effective density of the object the higher the X-ray attenuation and the darker it appears in the radiograph. The Invar 36 powder appears as light grey, whilst the molten pool and melt track are both dark grey, since they are almost twice as dense as the powder and hence attenuate more X-rays.”*

[Line 82 – 84] *“During the initial stages of LAM (**Fig. 1b**), a molten pool forms ca. 100 μm (twice the laser spot size) below the powder bed surface ($t = 1.8 \text{ ms}$) and rapidly grows into a 250 μm sphere ($t = 2.8 \text{ ms}$).”*

What is the definition of the “denuded zone”?

Response: We have clarified that the denuded zone is a powder free zone created by the laser-induced metal vapour/gas jet:

[Line 84 - 87]: *“...powder spatter removes a significant amount of powder particles ahead of the laser beam, forms a powder-free zone (or denuded zone) normal to the*

melt surface, so that there are fewer powder particles available for subsequent powder consolidation.”

Why have the authors not considered re-deposition of spatter and droplets as a key aspect of the process? The images at 384 and 390ms seem to suggest this possibility.

Response: In this figure, there are powder particles adhered to the walls which look like spatter above the pool. However, as detailed in the text, the laser beam was switched off at 334. Therefore, it was not possible to form spatter between 384 and 390 ms. During this period, we could only see the moving pores in the melt track. We have made this statement clearer in the revised manuscript.

[Line 116 – 117] *“The melt track continues to grow until the laser switches off at 334 ms. At this time, no porosity is evident in the melt track and spattering has also stopped...”*

7. Line 87: “The powder particles are often entrained by the gas/vapour jet”: How does one determine whether or not powder particle is entrained?

Response: Our statement is based on the greyscale of the images, because powder particles attenuate fewer X-rays than the molten pool. Based on the resolution of the image and the greyscale of the object, we can distinguish the difference between powder particles and large molten droplets. In the revised manuscript, we have rephrased the paragraph to address the resolution of the imaging system.

[Line 101 - 104] *“We can distinguish whether spatter is hot or cold (Ly et al. 2017) by its greyscale values and diameter. The hot ejection observed usually has a diameter larger than 40 μm (> 5 pixels); however, we also observed cold powder agglomerates rather than individual particles(Ly et al. 2017) being ejected (recognisable by its low greyscale value compared to that of fully dense hot ejection)..”*

8. Line 106: If powder spatter is remelted by the laser beam, why wouldn't individual powder particles also be remelted? Is this a resolution issue? The authors should comment on resolution issues somewhere in the main body of the paper.

Response: The reviewer is correct, our conclusions are limited by the spatial resolution of the imaging system. This has been made more explicit in the text. However, we are able to resolve that powder spatter occurs both as individual particles and as agglomerates. However, we cannot resolve the melting of individual powder particles since the density of the solid and liquid is very similar, and hence attenuation variation is minimal.

[Line 101 - 104] *“We can distinguish whether spatter is hot or cold (Ly et al. 2017) by its greyscale values and diameter. The hot ejection observed usually has a diameter larger than 40 μm (> 5 pixels); however, we also observed cold powder agglomerates rather than individual particles(Ly et al. 2017) being ejected (recognisable by its low greyscale value compared to that of fully dense hot ejection)..”*

9. Figure 2: Clearly identify scan direction and include a scale on the times in the figure.

Response: We have added the scan direction of the laser beam and reformatted the time colour bar in **Fig. 2** to make it clearer, therefore we have also amended the caption by adding a description of the red arrows.

[Fig. 2 – caption] “...The black dotted line indicates the powder bed surface. Red arrows indicate the scan direction of the laser beam. Scale bars, 2 mm....”

10. Figure 3 – Explain the contrast and shape of MT1 (why is there a dark ellipsoidal shape for MT1?), how is it clear that MT2 is liquid in the first frame?

Response: **Fig. 1a** at 400 ms shows the final state of MT1, wherein the dark ellipsoidal objects are metal beads formed ahead of MT1.

Fig. 3a shows that a molten pool was first formed on loose powder (highlighted by the orange circle). At 0.8 ms, powder consolidated into a molten pool, increased the density of the object, hence the molten pool attenuates more X-rays than the powder particles and it appeared a dark grey. At 4.8 ms, the molten pool grew into its

optimum size (**Fig. 3e**) and reached a density closed to MT1. Hence, the greyscale of the molten pool and MT1 were very similar.

Why is there an abrupt change in contrast in MT2 at the centre in frame b?

Response: The abrupt contrast at the centre of **Fig. 3b** shows that a powder layer was trapped between MT1 and MT2 during LAM which has been depicted in the text. We have made this statement more explicit in the next version of the main text and figure caption.

[Line 209 - 212]: “The Marangoni-driven flow also entrains a thin powder layer between MT1 and MT2 indicated by the abrupt changes in the image contrast at the centre of the red dotted circle in Fig. 3h.”

Where is the purple line in k?

Response: The purple circles highlight the droplet spatter in **Fig. 3c-d**, we have altered the **Fig. 3**'s caption to make this clearer, see the highlighted text.

[Fig. 3's Caption] “**Fig. 3: Time-series radiographs acquired during LAM of an Invar 36 second layer melt track (MT2) under $P = 209$ W, $v = 13$ mm/s and $E_L = 16.1$ J/mm, see details in Supplementary Movie 5. Snapshots of MT2 at various stages of LAM: a, Formation of a molten pool on the second powder layer, b, growth of MT2 via wetting on MT1, c, MT2 undergoes laser-re-melting, where RZ indicates the laser re-melting zone; d. MT2 completely solidifies. (c-d) The large droplet spatter (purple dotted circle) failed to eject from the powder bed.** Other key

phenomena involved during molten pool wetting: e, Re-melting of MT1; f, extension of MT2 before wetting on MT1; g, entrapment of gas bubble (see yellow dotted circles) and h, thin powder layer (see red dotted circles). Three new pore evolution mechanisms induced by laser re-melting: i, pore coalescence, j, pore dissolution and k. dispersion into smaller pores. Scale bars, 500 μm .”

11. Line 225: During the formation of MT2, the laser strikes the top surface of MT1 rather than the powder surface, with a greatly reduced in surface area, creating a gas/vapour jet that is only sufficient to eject powder particles.” The authors are suggesting that the powder particles absorb no energy in MT2? This seems unlikely. Also, there is no estimate of the fraction of energy absorbed in the MT1 melting configuration as compared to MT2.

Response: We have rephrased this paragraph to clarify MT2 forms less spatter than MT1.

[Line 248 – 250] *“During the formation of MT2, the laser beam fuses the powder layer with MT1 so that much of the heat energy is transferred to MT1, and hence only a small amount of energy contributes to gas/vapour jet and spatter formation.”*

In terms of the fraction of energy that is converted to form a melt track, this can indeed be estimated by the volume of melt track formed in each case, and is a good suggestion for future work.

12. The observation of different porosity formation mechanisms in MT2 is particularly important. The assertions (Line 192) about thermal gradients, surface energy gradients and Marangoni flow should be clarified – to what extent are there observations/simulations to support this and to what extent are the authors hypothesizing?

Response: Without the presence of surfactants, the direction of the Marangoni flow for most metals/alloys followed an inward Marangoni flow owing to their negative temperature dependent surface tension coefficient (Valencia and Queded 2008). By tracking the pore flow pattern, we can infer that the pore motion follows an inward Marangoni convection, suggesting that the surface tension is low at the laser-matter interaction zone compared to the edges of the melt track. We believe that these observations were sufficient to explain the Marangoni flow with respect to thermal gradients and surface tension.

13. Section 5: What is the minimum and maximum size of spatters being tracked in the trajectory analysis? Again, redeposition should be addressed in the context of beam velocity and redeposition time; is it just at a length scale and time scale beyond what is observed here?

Response: Based on the spatial resolution of the X-ray images, we were able to track powder / droplet spatter with a size ranging from 40 μm to 350 μm . We have clarified the spatial resolution of the X-ray imaging setup in the revised manuscript [Line 102].

[Line 243 - 244] *“Due to the spatial resolution of the radiographs, we are only able to track spatter with a size ranging of 40 – 350 μm .”*

If I understand the reviewer’s question correctly, we did not observe the redeposition “of powder or droplet spatter” on the melt track possibly due to the spatial resolution of the experimental setup. However, no melt beads or powder particles were found attached to the top surface of the melt track in the SEM images, suggesting most spatter objects were ejected far away from the scanned track.

14. Are there any sights on final surface roughness related to the phenomena observed?

Response: Yes, we observed that the resulting melt features exhibited smooth top surfaces. A sintered powder layer usually attached to the bottom of the melt features, which increases the surface roughness. However, our study focuses on the transient phenomena that occurred during LAM rather than on the surface roughness of the melt features. Nevertheless, we have taken this comment on board and we may look at this topic in a separate paper.

15. Finally, the numerous grammatical errors in the manuscript lead the reviewer to question whether the two senior authors have thoroughly enough reviewed this manuscript. Some examples include:

Response: We have corrected the following grammatical errors, and done a thorough proof read of the final manuscript.

Line 27: “Laser additive manufacturing....have attracted”

Line 28: “build ups”

Line 34: “To solve these challenges require a better understanding....”

Line 37: “not well explained owing to complex molten pool behaviour occurs on very short time scales (10⁻⁶ - 10⁻³ s).

Line 42: “the development of LAM and simulation tools required to apply on new materials”

Line 76: “because spatter removes a large amount of materials”

Line 82: “Notably, the hydrodynamic movement of the melt fluid du ring wetting combined with the differential vapour pressure at the laser-powder interaction zone, causing the molten pool to oscillate throughout LAM, a similar prediction is reported in selective electron beam melting.”

Line 94: “The existing hypothesis suggests “...what is the hypothesis?”

Response: The existing hypothesis refers to the reference articles by (Liu *et al.* 2015, Matthews *et al.* 2016, Bidare *et al.* 2018). In order to make this clearer, we have rephrased the paragraph.

[line 92 – 101] *“The intense laser beam with a power density of $\sim 10^6$ W/cm² causes metal vaporisation at the molten pool surface and generates a recoil pressure in the laser-matter interaction zone, resulting in a gas/vapour jet. We hypothesise that the metal vapours heat the argon gas and cause a gas expansion, resulting in a vapour jet that expands radially at high-speed. This laser induced gas/vapour jet entrains powder particles into a melt track which is a key track growth mechanism (blue dotted circles in Fig. 1d). It induces powder spatter from the laser-matter interaction*

zone (blue dotted circles, Fig. 1e) while forming a denuded zone. Unlike the droplet spatter mechanisms reported in reference articles (Liu et al. 2015, Matthews et al. 2016, Bidare et al. 2018), we reveal that the majority of the droplet spatter during LAM in overhang condition are originated from molten pools ahead of the melt track or melting of powder spatter by the laser Ly et al, (Supplementary Fig. 4b).

Reviewer #4 (Remarks to the Author):

Excellent, original and comprehensive study of the physics of the selective laser melting process. It clearly elucidates the physical processes occurring during laser melting of powder materials under a range of operating conditions.

Response: We'd like to thank reviewer #4 for your compliment on our work.

I was wondering if you can clarify what effect does powder compaction play in the process. In your experiments the powder was loosely spread on the substrate while in practice its packing density is greater which can impact the process.

Response: Increased powder compaction would certainly increase the effective absorption and thermal conductivity of the powder bed, increasing the efficiency of the melting process. When the powder compaction approaches to 100%, then we would be processing a solid plate rather than powder bed, in which case the molten pool dynamics would be similar to that for laser welding.

The design of our experiment mimics a commercial powder bed fusion system where the powder packing density is ca. 40 – 60%, depending on the powder size distribution. This has been further clarified in the text:

[line 55 - 56] *“The LAMPR mimics an entry-level commercial LAM system, including a laser system, a loose powder bed with a packing density of 40 – 60% and an inert processing environment.”*

Also have found numerous grammatical expressions which should be corrected before publication.

Response: We hope we have corrected all the grammatical errors in the paper including those listed below. We apologise for these errors.

Line 37 Include "that" after "behaviour"

Line 39 Change "These" to "This"

Line 76 and 77, I would change "material" to "powder"

Line 83 Change "causing" to "causes"

Line 89 Sentence starting with "Similar ..." needs to be looked at.

Line 96 Change "attributes" to "contributes"

Line 121 Change "they" to "They" after "."

Line 213 Figure 3. Purple dotted circles not included on my copy of the manuscript.

Line 226 Delete "in" after "reduced"

Line 230 Change "confirms" to "confirming"

References

- Karagadde, S, Lee, PD, Cai, B, Fife, J.L., Azeem, M.A., Kareh, K.M., Puncreobutr, C., Tsivoulas, D., Connolley, T. & Atwood, R.C., "Transgranular liquation cracking of grains in the semi-solid state", *Nature Commun.*, 5:9300, 10.1038/ncomms9300, 2015.
- Kawahito, Y, Mizutani, M and Katayama, S, "Elucidation of high-power fibre laser welding phenomena of stainless steel and effect of factors on weld geometry", *J. Phys. D: Appl. Phys.* 40 (2007) 5854–5859
- Lee, P.D. and Hunt, J.D., "Hydrogen Porosity in Directionally Solidified Aluminium-Copper Alloys: In-Situ Observation", *Acta Mat.* 45 (10), 4155-4169, 1997. /10.1016/S1359-6454(97)00081-5
- Liu, Y., Yang, Y., Mai, S., Wang, D. and Song, C. Investigation into spatter behaviour during selective laser melting of AISI 316L stainless steel powder. *Materials and Design* 87, 797–806 (2015).
- Matthews, M. J., Guss, G., Khairallah, S. A., Rubenchik, A. M., Depond, P. J. and King, W. E. Denudation of metal powder layers in laser powder bed fusion processes. *Acta Materialia* 114, 33–42 (2016).
- Ly, S., Rubenchik, A. M., Khairallah, S. A., Guss, G. & Matthews, M. J. Metal vapor micro-jet controls material redistribution in laser powder bed fusion additive manufacturing. *Sci. Rep.* 7, 4085 (2017).
- Bidare, P., Bitharas, I., Ward, R. M., Attallah, M. M. and Moore, A. J. Fluid and particle dynamics in laser powder bed fusion. *Acta Materialia* 142, 107–120 (2018).
- Wang, J., Li, M, Allison, J., and Lee, P.D., "Multiscale Modeling of the Influence of Fe Content in a Al-Si-Cu Alloy on the Size Distribution of Intermetallic Phases and Micropores", *J. App. Phy.*, 107, 061804, 2010. /10.1063/1.3340520
- Valencia, J. J. and Quested, P. N. Thermophysical properties. *ASM Handbook: Casting* 15, 468–481 (2008).

Reviewers' Comments:

Reviewer #2:

Remarks to the Author:

The author response to review, while thorough, fails to address my basic concerns related to how relevant the reported measurements are to the actual process of LPBF. The authors arguments that interaction with BN windows does not significantly effect welding studies does not necessarily mean that window interaction has negligible effects on a powder bed. It is well established that powder motion in LPBF is extensive and important to melt pool physics, which is a significant contrast to the much simpler case of welding. Furthermore, the scan speeds used in this study are two orders of magnitude slower than those used in a typical LPBF build for this material (see Qui et al, Acta Materialia, 2016, 103, 382-395). Such different scan speeds likely place the phenomena observed here in a different processing regime than a typical build, because they are so far outside the normal process window. Typical approaches to robust overhang regions use lower power and the same scan speed, not orders-of-magnitude slower scanning. The authors do acknowledge these issues, however, and the observations related to Maragoni convection and pore formation and motion are quite interesting and valuable contributions. The authors should address the below minor revisions.

Specific comments:

Line 52: should be "operando" rather than "in operando"

Line 84: the 500 um diameter sphere described here cannot be a sphere, as it is constrained to 300 um in one dimension. Is there evidence of a change of growth rate at around 300 um as a result of this constraint?

Line 89: is this wetting to the substrate walls, or new powder?

Line 102: It is not clear what is meant by "molten pools in front of the melt track" in this context. The authors claim the spatter they see in this case arises from a different mechanism, but it is not obvious how that mechanism is different than what has been seen before in other studies.

Previous reports discuss spatter that is ejected from the front of the melt pool (see references 40, 41)

Line 135: Figure 1 is a bit confusing. Using letters multiple times, in some cases in at least 3 places in the figure, initially made the figure difficult to understand. The cartoons do not add much - you may be better served by making the actual data panels bigger so they are easier to see/label and either eliminating or shrinking the cartoon portions, which are rather redundant to the data.

Line 149: comparing linear energy density between different works should be done with caution, as it depends heavily on beam size.

Line 240: Is it possible that some of the gas in the pores that you believe to contain nitrogen are from dissociation of boron and nitrogen in BN? The metal vapor which comes in contact with the BN walls may be hot enough to cause some dissociation

Line 267: Figure 4b would be better if the x and y location axes were in micron rather than pixels

Reviewer #3:

Remarks to the Author:

The authors have sufficiently addressed the concerns of the reviewers and the manuscript should be published.

Reviewer #4:

Remarks to the Author:

You have significantly improved the paper and have addressed fully my comments. I recommend that the paper be published as is.

Response to review #2:

We are grateful for the referee's revised comments and have detailed our responses below (in purple text) where we have tried to take on board the full spirit of the remarks.

Reviewer #2 (Remarks to the Author):

The author response to review, while thorough, fails to address my basic concerns related to how relevant the reported measurements are to the actual process of LPBF. The authors arguments that interaction with BN windows does not significantly effect welding studies does not necessarily mean that window interaction has negligible effects on a powder bed. It is well established that powder motion in LPBF is extensive and important to melt pool physics, which is a significant contrast to the much simpler case of welding. Furthermore, the scan speeds used in this study are two orders of magnitude slower than those used in a typical LPBF build for this material (see Qiu et al, Acta Materialia, 2016, 103, 382-395). Such different scan speeds likely place the phenomena observed here in a different processing regime than a typical build, because they are so far outside the normal process window. Typical approaches to robust overhang regions use lower power and the same scan speed, not orders-of-magnitude slower scanning. The authors do acknowledge these issues, however, and the observations related to Maragoni convection and pore formation and motion are quite interesting and valuable contributions. The authors should address the below minor revisions.

Response: We appreciate Reviewer#2 comments questioning whether the processing regime we used is comparable to the industrial conditions as used by Qiu *et al*.

[line 182-189]: The large difference in LED, because the effective thermal conductivity of the powder bed (present study) is lower than that of building on a solid starting block.¹ In addition, the wettability of the molten pool is significantly lower when building on powder support than building on a solid starting block or a prior solid track.² For both reasons, a slower scan speed is required to form a continuous track on powder support as compared to solid; therefore, requiring us to use different processing conditions to those used by Qiu et al..³ The mechanism map (Fig. 2) reveals morphological transitions of the melt track similar to reference studies^{4,3,2}, suggesting that the processing regime we selected is reasonable for laser melting directly on powder support.

To ensure no confusion we have now made this point very clearly in the paper with reference papers^{1,2}.

Specific comments:

Line 52: should be "operando" rather than "in operando"

Response: We have corrected that in the main text.

Line 84: the 500 μm diameter sphere described here cannot be a sphere, as it is constrained to 300 μm in one dimension. Is there evidence of a change of growth rate at around 300 μm as a result of this constraint?

Response: The molten pool can exhibit a spherical shape up to 300 μm . We agree with Reviewer #2 that the molten pool is constrained by the boron nitride walls once it reaches above 300 μm and have further clarified in the text, more clearly stating equivalent diameter was used.

To further prove the conclusions are correct, we selected two extra datasets to confirm that there are no changes in growth rate after the molten pool/melt track made contact with the BN walls.

Line 89: is this wetting to the substrate walls, or new powder?

Response: The newly formed molten pool wets onto the previous melt feature, see line 87 – 89 in the original text.

[Line 87-89]: “Consequently, the newly formed molten pool coalesces with the first melt bead via wetting ($t = 7.4$ ms), revealing a key track growth mechanism.”

Line 102: It is not clear what is meant by "molten pools in front of the melt track" in this context. The authors claim the spatter they see in this case arises from a different mechanism, but it is not obvious how that mechanism is different than what has been seen before in other studies. Previous reports discuss spatter that is ejected from the front of the melt pool (see references 40, 41).

Response: All previous work has been on laser additive manufacturing of powder on substrate. Here, we study the overhang build and layer by layer conditions, also reveal how the formation of the denuded zone forces the molten pools to be formed ahead and above of the melt track (see Fig. 1c) promoting spatter formation as LAM progresses. Further, we demonstrate that the amount of spatter increases as the molten pool develops into a melt track, and that powder spatter can form from unmelted powder agglomerate and transforms into a droplet spatter as it passes through the laser beam – mechanisms that have not been reported before. Our observations therefore both support, and expand, upon those of Ly et al. We have further clarified in the text.

[line 102 -106] “Hot ejection has a diameter greater than 40 μm (or ~ 6 pixels) and it appears as black in the X-ray images as compared to cold powder agglomerates (which are less dense than the hot ejection and hence a dark grey colour in the X-ray images). Our results reveal that, in addition to the individual particle ejection⁵, cold powder agglomerates are also ejected, interacting with the laser beam to become molten, transforming cold ejection into hot spatter. Although neither studies have a direct temperature measurement, our results support and build on the hypothesis of Ly et al.⁵ on the formation of cold and hot ejections during LAM.

Line 135: Figure 1 is a bit confusing. Using letters multiple times, in some cases in at least 3 places in the figure, initially made the figure difficult to understand. The cartoons do not add much - you may be better served by making the actual data panels bigger so they are easier to see/label and either eliminating or shrinking the cartoon portions, which are rather redundant to the data.

Response: We have removed Figure 1g and enlarged Figure 1b-f. In addition, we have also amended labels, caption and the main text according to the new Figure 1.

Line 149: comparing linear energy density between different works should be done with caution, as it depends heavily on beam size.

Response: We agree, which is why we carefully measured beam diameter and report it.

Line 240: Is it possible that some of the gas in the pores that you believe to contain nitrogen are from dissociation of boron and nitrogen in BN? The metal vapor which comes in contact with the BN walls may be hot enough to cause some dissociation

Response: The decomposition of BN into boron and nitrogen in an inert atmosphere requires a temperature of $ca. 2000 K^6$. After the experiment, the BN is non-wetted with Invar 36. As the reviewer suggests, it is unlikely that there is any decomposition of BN.

Line 267: Figure 4b would be better if the x and y location axes were in micron rather than pixels.

Response: Thank you for the suggestion. We have made changes to Figure 4b.

References:

1. Gusarov, A. V., Laoui, T., Froyen, L. & Titov, V. I. Contact thermal conductivity of a powder bed in selective laser sintering. *Int. J. Heat Mass Transf.* **46**, 1103–1109 (2003).
2. Li, R., Liu, J., Shi, Y., Wang, L. & Jiang, W. Balling behavior of stainless steel and nickel powder during selective laser melting process. *Int. J. Adv. Manuf. Technol.* **59**, 1025–1035 (2012).
3. Qiu, C., Adkins, N. J. E. & Attallah, M. M. Selective laser melting of Invar 36: Microstructure and properties. *Acta Mater.* **103**, 382–395 (2016).
4. Yadroitsev, I., Gusarov, A., Yadroitsava, I. & Smurov, I. Single track formation in selective laser melting of metal powders. *J. Mater. Process. Technol.* **210**, 1624–1631 (2010).
5. Ly, S., Rubenchik, A. M., Khairallah, S. A., Guss, G. & Matthews, M. J. Metal vapor micro-jet controls material redistribution in laser powder bed fusion additive manufacturing. *Sci. Rep.* **7**, 4085 (2017).
6. Dreger, L. H., Dadape, V. V. & Margrave, J. L. SUBLIMATION AND DECOMPOSITION STUDIES ON BORON NITRIDE AND ALUMINUM NITRIDE. *J. Phys. Chem.* **66**, 1556–1559 (1962).